# Timing of Fungal Insecticide Application to Avoid Solar Ultraviolet Irradiation Enhances Field Control of Rice Planthoppers

**DOI:** 10.3390/insects14040307

**Published:** 2023-03-23

**Authors:** Wan-Ying Xu, Zhen-Xin Wen, Xin-Jie Li, En-Ze Hu, Dan-Yi Qi, Ming-Guang Feng, Sen-Miao Tong

**Affiliations:** 1College of Advanced Agricultural Sciences, Zhejiang A&F University, Hangzhou 311300, China; 2Jixian Honors College, Zhejiang A&F University, Hangzhou 311300, China; 3Institute of Microbiology, College of Life Sciences, Zhejiang University, Hangzhou 310058, China

**Keywords:** *Beauveria bassiana*, *Metarizhium anisoplae*, optimal application strategy, rice insect pests, biological control

## Abstract

**Simple Summary:**

The control of rice planthoppers (RPH) in Asian countries has relied upon chemical insecticides for several decades. The reliance no longer continues in the rice–aquaculture coculture and rotation systems emerging as high-profit agriculture in Southern China because chemical control is prohibited in these systems to prevent aquaculture products from contamination with chemical residues. In this field study, the RPH population in three rice–shrimp rotation paddy fields was effectively controlled for four weeks at the critical tillering to flowing stages after two fungal insecticides were sprayed twice at the recommended rate. For either fungal insecticide, the sprays applied after 5:00 p.m. to avoid solar UV irradiation were consistently more efficacious against the RPH population than those before 10:00 a.m. These results indicate the feasibility of fungal insecticides for RPH control in the rice–shrimp rotation fields and the importance of timing the application of fungal insecticides to avoid solar UV exposure for enhanced RPH control during the summer months.

**Abstract:**

Thechemical control of rice planthoppers (RPH)is prohibited in annual rice–shrimp rotation paddy fields. Here, the fungal insecticides *Beauveria bassiana* ZJU435 and *Metarizhium anisoplae* CQ421 were tested for control of RPH populations dominated by *Nilaparvata lugens* in three field trials. During four-week field trials initiated from the harsh weather of high temperatures and strong sunlight, the rice crop at the stages from tillering to flowering was effectively protected by fungal sprays applied at 14-day intervals. The sprays of either fungal insecticide after 5:00 p.m. (solar UV avoidance) suppressed the RPH population better than those before 10 a.m. The ZJU435 and CQ421 sprays for UV avoidance versus UV exposure resulted in mean control efficacies of 60% and 56% versus 41% and 45% on day 7, 77% and 78% versus 63% and 67% on day 14, 84% and 82% versus 80% and 79% on day 21, and 84% and 81% versus 79% and 75 on day 28, respectively. These results indicate that fungal insecticides can control RPH in the rice–shrimp rotation fields and offer a novel insight into the significance of solar-UV-avoiding fungal application for improved pest control during sunny summers.

## 1. Introduction

Rice planthoppers (RPH), including mainly *Nilaparvata lugens* (Stål) and *Sogatella furcifera* (Horvath), are notorious sap-sucking pests that have severely threatened rice production since the 1980s. Frequent outbreaks of RPH populations are ascribed to seasonal migration from south to north and vice versa in rice-growing countries of Eastern Asia [1,2,3,4,5], rapid development of resistance to various chemical insecticides [6,7,8,9,10], resurgence induced by overuse of pesticides [11], and adaptation to resistant rice varieties [12,13]. Air temperature and CO_2_ concentration elevated due to climate change facilitate RPH outbreaks [14,15,16,17]. Conventional fertilizer application necessary for rice growth also promote RPH reproduction [18,19]. While polycultural measures and resistant rice varieties are pursued [20,21,22], RPH control remains dependent on chemical insecticides despite the increasing development of resistance. It is, therefore, a great challenge to develop alternative strategies for effective RPH control.

High-profit agriculture has been emerging in Southern China, expanding from traditional rice–fish coculture systems [23,24] to rice–shrimp, rice–crab, rice–soft shell turtle and rice–bullfrog coculture or rotation systems [25,26,27]. To prevent the high-profit aquacultures from contamination with chemical residues, insecticidal compounds are prohibited for field control of major rice pests, such as RPH and leaf folder pests that cause damage in the coculture and rotation systems. Such a demand appears to favor environment-friendly fungal insecticides, whose active ingredients are the formulated conidia of insect-pathogenic fungi, such as *Beauveria bassiana* and *Metarhizium anisopliae* serving as the active ingredients of wide-spectrum mycoinsecticides [28]. Previously, emulsifiable oil formulation of either *B. bassiana* or *M. anisopliae* showed high potential for RPH control under laboratory and field conditions [29,30]. Application of *B. bassiana* granules to the water surface in contact with rice plants led to as effective an RPH control as a spray of conidia under controlled conditions [31]. Field trials with a formulation of *M. anisopliae* CQ421 showed a RPH control efficacy of 50–70% on day 7 after spraying and little impact on native microbiota [32,33]. The same formulation has also proved effective for the sustainable control of major rice insect pests, including leaf folders and stem borers as well as RPH, with minimal effects on natural enemies of those pests, in multiple provinces of China during 2011–2018 [34]. The previous studies suggest that conidial formulations of *B. bassiana* and *M. anisopliae* serve as desirable alternatives for the control of insect pests in rice–aquaculture systems.

Summer, the main season of rice pest infestation, features high temperatures and sunlight that limit the application of fungal insecticides [35,36,37,38]. UVB, a major solar ultraviolet (UV) component (280–320 nm) in sunlight, is very harmful to formulated conidia [39] and, hence, must be avoided when a fungal insecticide is applied, but this was not addressed in the previous field studies. Based on daily accumulation patterns of solar UVB dose on sunny summer days, fungal tolerance to a UVB dose of at most 0.5 J/cm^2^ and fungal ability to photorepair UVB-induced DNA lesions for photoreactivation of UVB-impaired conidia, low-risk and no-risk strategies have been proposed for fungal insecticides to be applied between 3:00 p.m. and 5:00 p.m. and after 5:00 p.m. [40], respectively. Application at 3:00–5:00 p.m. will involve an accumulated UVB dose of ~0.2 J/cm^2^, which causes moderate fungal damage that can be self-photorepaired. There is little UVB accumulation after 5:00 p.m. even on sunny days. The proposed strategy recognizes a high risk for fungal spray in the morning of a sunny day since a UVB dose accumulated from 9:00 a.m. to 3:00 p.m. greatly exceeds the upper limit of fungal tolerance.

This study seeks to test the efficacies of two typical mycoinsecticides applied before 10:00 a.m. and after 5:00 p.m. for the control of insect pests in the rice crop of an annual rice–shrimp rotation system. Our goals are as follows: (1) determining whether wide-spectrum fungal insecticides are practically feasible for the management of major rice insect pests in the rotation system; (2) comparing the control efficacies of *B. bassiana* and *M. anisopliae* during a four-week period of summer; and (3) testing whether fungal application after 5:00 p.m. to avoid solar UV exposure is more efficacious than that in the morning for exposure to solar UV. This paper focuses on the presentation of fungal control efficacies against RPH populations.

## 2. Materials and Methods

### 2.1. Fungal Formulations

The fungal strains *B. bassiana* ZJU435 (China General Microbiological Culture Collection Center, CGMCC No. 13566; designated Bb) and *M. anisopliae* CQ421 (CGMCC No. 460; designated Ma) were officially registered as wide-spectrum fungal insecticides in China and were used in this field study. An emulsifiable oil formulation of either Bb (1 × 10^10^ conidia/mL) or Ma (8 × 10^9^ conidia/mL) was provided by the manufacturer Greenation (Chongqing, China).

### 2.2. Rice–Shrimp Rotation Paddy Fields

Three rice–shrimp rotation paddy fields (0.5–0.6 ha each) were located in Lianfeng Village (30°33′39.468″ N 120°24′8.133″ E), Chongfu Town, Tongxian City, Zhejiang Province for arrangement of three repeated field trials (Trials 1–3). The rice–shrimp rotation usually starts from mechanic direct seeding of rice in mid-June. The rice crop is harvested in late October, followed by immediate soil turnover for the cultivation of water algae and grasses, the release of shrimp (*Macrobrachium rosenbergii*) larvae into the field in January, and the marketing of the aquaculture from late March to early June. After the final shrimp harvest, soil turnover is carried out again for the initiation of the rice crop. For this study, the direct seeding of rice (*Oryza sativa* var. Jiahe 218) in the three fields was conducted on 17 June 2022. Field trials were established at the stage of tillering peak on 10 or 11 August.

### 2.3. Field Trials

Each of the fields was divided into 15 plots [176 m^2^ (8.8 m × 20 m) per capita] with 3 m wide edge buffers of the whole field and a 2 m wide buffer between plots. Each trial included four fungal treatments and one blank control shared by the fungal treatments; either Bb or Ma formulation was applied to three plots before 10:00 a.m. (used as solar UV exposure (UVexp) treatment) or after 5:00 p.m. (used as solar UV avoidance (UVavo) treatment). Three plots were used as blank control. All fungal treatments and blank control in each field trial were randomly arranged.

For the UVexp and UVavo treatments, each plot was treated at a rate of 23.3 mL of Bb (1.32 × 10^13^ conidia/ha) on 10 August and of 26.7 mL of Ma (1.21 × 10^13^ conidia/ha) on 11 August. Either 23.3 mL of Bb or 26.7 mL of Ma formulation was suspended in 10 L of water (i.e., 429- or 375-fold dilution). The suspension was sprayed using a knapsack airblast sprayer driven by a 12 V lithium battery at a power of 25 W (Lanyi ScienTech Co., Shengzhen, China). The same rates of Bb and Ma were repeatedly applied on August 24 and 25, respectively. Each plot of the control was sprayed with the same volume of water.

To examine the uniformity of the fungal spray, double-side adhesive tape was used to fix a cover slip onto each of three rice plants at 5 m intervals in each fungal treatment sprayed on August 10 or 11 to collect the deposited conidia. All cover slips collected after spraying were individually mounted on slides and carried back to the laboratory for microscopic examination. Counts of conidia were made from four fields of view (*ϕ* = 0.52 mm) of each cover slip and converted to the number of conidia deposited per unit area (mm^2^) of the rice plant surface.

### 2.4. Sampling for RPH Population and Control Efficacy

The RPH population was dominated by *N. lugens* over the four-week period of field trials and monitored weekly by taking 4 samples diagonally at 4 m intervals in each plot of either fungal treatment. For the blank control shared by two fungal insecticides, 4 samples were taken diagonally at opposite directions in each of three plots on the sampling occasions of Bb (Control 1) and Ma (Control 2, one-day later than Control 1). At each of the sampling sites, the RPH density was estimated by patting and shaking the rice plants in a quadrat of 0.25 m^2^ to release all the active adults and nymphs onto a white tray for counting in situ. On each sampling day, the sampling was completed early in the morning by making use of dews, enabling the wetting of the dropped hoppers for making the counting in situ easier.

The relative control efficacy (%) of each fungal treatment (UVAv or UVEx) in each field trial was calculated using the formula [1 − (*D*_tj_*D*_c0_/*D*_t0_*D*_cj_)] × 100. In the formula, *D*_c0_ and *D*_cj_ are the respective RHP densities estimated in the corresponding blank control on the initial day (10 or 11 August) and the *j*th day after the first spray; *D*_t0_ and *D*_tj_ denote the RHP densities estimated in each fungal treatment on the initial day and the *j*th day after the first spray, respectively. The first spray was conducted after sampling early in the morning of the same day.

### 2.5. Weather Data

Weather data associated with fungal insecticidal activity during the period of field trials were provided by the local weather station, including daily mean, maximal, and minimal temperatures, and daily precipitation (mm).

### 2.6. Statistical Analysis

A one-way analysis of variance (ANOVA) was carried out to reveal the variation among the counts of deposited conidia (normalized by conversion to logarithms) from the first spray and variations among the estimates of RPH densities (normalized by conversion to logarithms) and the relative control efficacies (normalized by conversion to arcsine square roots) among the fungal treatments on each sampling day in each field trial, followed by multiple comparison of means through Tukey’s honestly significant difference (HSD) test. The same analysis and Tukey’s test were also performed for differentiating the mean efficacies of all fungal treatments in three field trials on a given day after the first spray.

## 3. Results

### 3.1. Weather Conditions during Field Trials

The first week of field trials featured consistently high temperatures and strong sunlight (Figure 1). The mean and maximum of the daily temperature ranged, respectively, from 32.4 to 36.1 °C and 37.1 to 41 °C in the first week, making it necessary to consider the effect of solar UV damage on fungal insecticidal activity. Strong sunlight with a slightly lower temperature (daily means: 28.5 to 32.4 °C; daily maxima: 30.6 to 39.7 °C) continued in the second week. The following two weeks included six rainy days and featured frequent clouds and a daily mean temperature of 26.1 °C (±1.9), favoring fungal insecticidal activity.

### 3.2. Deposition Rates of Formulated Conidia after Spray

The counts of conidia deposited on the rice plant surface (Figure 2, Appendix A) showed no significant variation among three replicates in each trial (*p* > 0.05 in *F*_2,6_ test) or between the UVavo and UVexp treatments of either fungal insecticide (*p* > 0.05 in Tukey’s test), but a significant difference between Bb and Ma (*p* < 0.05 in Tukey’s test). No significant variation was found among the three field trials of Bb (*F*_5,10_ = 0.20, *p* = 0.95) or Ma (*F*_5,10_ = 0.32, *p* = 0.89), resulting in overall mean (±SD) Bb and Ma deposition rates of 415 (±23) and 325 (±24) conidia/mm^2^ of the rice plant surface (*n* = 18), respectively. The mean deposition rates suggested that smaller *B. bassiana* conidia were deposited more readily on to the rice plant surfaces than larger *M. anisopliae* conidia.

### 3.3. Suppression of RPH Population by Fungal Sprays

The first sampling, early in the mornings of August 10 and 11, resulted in mean (±SD) densities of 12.2 (±2.0) and 12.8 (±2.1) RPHs per sample (*n* = 72), respectively. The initial samples showed no significant variation among the treatments (*p* ≥ 0.08 in *F*_5,10_ test) and replicates (*p* ≥ 0.16 in *F*_2,10_ test) of each field trial. On day 7 after the first spray, the RPH density in each field trial decreased significantly in the fungal treatments (*p* < 0.05 in Tukey’s test) in comparison to blank controls (Figure 3, Appendix A). The decrease was significantly greater in the treatment of UVavo than of UVexp in all trials of Bb but only in Trial 2 of Ma. The sharply increased RPH density in the control on day 14 resulted in an insignificant difference between the two treatments of either fungal insecticide. The second spray provided after sampling early in the morning on day 14 consistently suppressed the RPH population to low levels for two more weeks, during which the significance of a difference between the UVavo and UVexp treatments was masked by higher RPH densities in the control.

### 3.4. Control Efficacies of Two Fungal Insecticides against RPH Population

The relative control efficacies computed with the RPH densities in the samples of each fungal treatment versus the blank control in all field trials are displayed in Figure 4 (listed in Appendix A). The efficacies estimated on day 7 after the first spray showed a large variation within each treatment of Bb or Ma. The variation diminished after the second spray of Bb on 24 August or Ma on 25 August. The one-way ANOVA on a given day after the first spray revealed higher efficacies in the treatment of UVavo than of UVexp. The efficacy differences between the two treatments on the two sampling days after the first spray were larger than the corresponding differences after the second spray on day 14.

The mean efficacies of Bb in three field trials were significantly higher in the treatment of UVavo than of UVexp (*p* < 0.05 in Tukey’s test on days 7, 14, 21, and 28) (Figure 5, Appendix A). In contrast, such a difference between the two treatments of Ma was significant only on days 14 and 28. The overall mean (±SD) efficacies in the UVavo versus UVexp treatments of Bb and Ma were 60.2% (±4.5) and 55.9% (±2.7) versus 40.6% (±4.7) and 44.8% (±8.5) on day 7; 77.0% (±1.3) and 77.8% (±1.7) versus 63.8% (±4.3) and 66.9% (±2.0) on day 14; 84.0% (±1.2) and 82.1% (±3.2) versus 79.6% (±0.7) and 78.9% (±1.9) on day 21; and 83.7% (±0.2) and 81.4% (±1.0) versus 78.9% (±1.4) and 74.6% (±3.1) on day 28, respectively. Importantly, no significant difference in RPH control efficacy was detected between the two fungal insecticides (*p* > 0.05 in Tukey’s test).

## 4. Discussion

The fungal application rates in previous field trials to control different insect pests spanned from 1 × 10^13^ to 1 × 10^14^ conidia/ha [41,42,43,44,45]. High application rates are too costly for pest control and, hence, are economically unfeasible. In this field study, we applied *B. bassiana* ZJU435 and *M. anisopliae* CQ421 at the respective rates of 1.32 × 10^13^ and 1.21 × 10^13^ conidia/ha, which were very close to the recommended lower limit and observed uniform deposition of conidia on rice plants at the stage of the tillering peak. The two fungal insecticides were proven to have similar RPH control efficacies in the field trials irrespective of higher or lower deposition rates on rice plants after spray. Different deposition rates leading to similar control efficacies suggest that oil-formulated conidia that land on the water surface infect *N. lugens* nymphs and adults inhabiting the lower parts of rice plants. This implication is seemingly in agreement with the previous report on similar RPH control efficacies achieved by directly spraying conidia onto rice plants and the application of fungal granules to the water-surface-contacting stem bases [31].

Over the 4-week period of field trials, the RPH population dominated by *N. lugens* was well controlled by either *B. bassiana* ZJU435 or *M. anisopliae* CQ421, sprayed twice at 14-day intervals. The first spray on 10 or 11 August was followed by two weeks of strong sunlight and high temperatures, contrasting with the other two-week weather changes that were obviously more suitable for fungal action after the second spray on 24 or 25 August. Despite the harsh weather conditions after the first spray, the RPH control efficacy was higher in the treatment of UVavo than of UVexp. This difference highlights the importance of fungal application to avoid solar UV damage. The nymphs and adults of *N. lugens* inhabit preferentially lower shaded parts of rice plants, which were presumably exposed to less UV and lower temperature than the upper parts. Perhaps for this reason, the first morning spray of either *B. bassiana* ZJU435 or *M. anisopliae* CQ421 followed by the harsh weather also resulted in significant RPH control despite lower efficacies (41–45%) on day 7 than those (56–60%) attributed to the spray after 5:00 p.m. The overall mean efficacy of RPH control by either fungal insecticide in three field trials was significantly higher in the treatment of UVavo (~77%) than of UVexp (~65%) on day 14 after the first spray. These observations were in agreement with the RPH control efficacy of 50–70% observed in the previous field trials of *M. anisopliae* CQ421 under undisclosed weather conditions [32,33]. The repeated spray of each fungal insecticide provided a consistent RPH control for two more weeks. Notably, while the weather changes during the period of field trials revealed an impact of high temperatures on the fungal efficacies against the RPH population, it is still unclear whether possible changes in the CO_2_ concentration (absent in the weather records) in the paddy fields affected the fungal efficacies during the period.

In conclusion, two sprays of either *B. bassiana* or *M. anisopliae* formulation at 14-day intervals resulted in desirable RPH control over the 4-week period from the tillering peak to the flowering stages. Application after 5:00 p.m. to avoid solar UV damage was proven to be more efficacious against the RPH population than the morning spray exposed to solar UV damage more accumulated in the daytime of sunny days [39]. These results confirm that *B. bassiana* and *M. anisopliae* formulations are alternative products to protect the rice crop in rice–shrimp rotation fields from RPH damage and likely in other rice– aquaculture systems where the application of chemical pesticides is prohibited.

## Figures and Tables

**Figure 1 insects-14-00307-f001:**
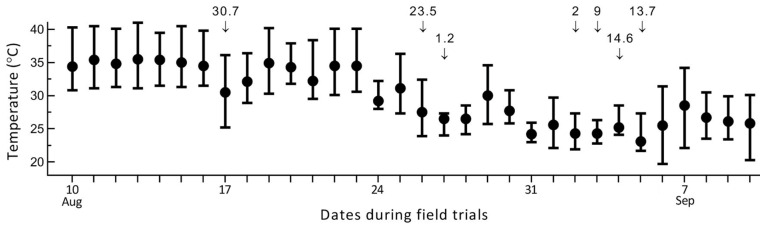
Mean (solid circle), maximum, and minimum (upper and lower limits) of daily temperature and daily precipitation (mm, arrowed) records over the four-week period of field trials after fungal application.

**Figure 2 insects-14-00307-f002:**
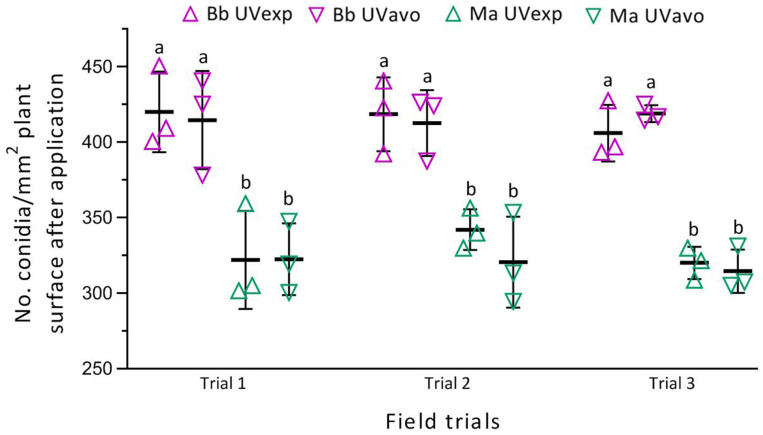
Deposition rates of emulsifiable oil-formulated *B. bassiana* ZJU435 (Bb) and *M. anisopliae* CQ421 (Ma) conidia. Bb and Ma were suspended in water (429- and 375-fold dilutions) and sprayed on the rice crop of tillering peak at the respective rates of 1.32 × 10^13^ (on 10 August) and 1.21 × 10^13^ conidia/ha (on 11 August). Deposition rates were log_10_-transformed in one-way ANOVA. Different lowercase letters indicate a significant difference (*p* < 0.05 in Tukey’s test) in each field trial. Error bars: standard deviations (SDs) of the means from three replicates (shown with symbols).

**Figure 3 insects-14-00307-f003:**
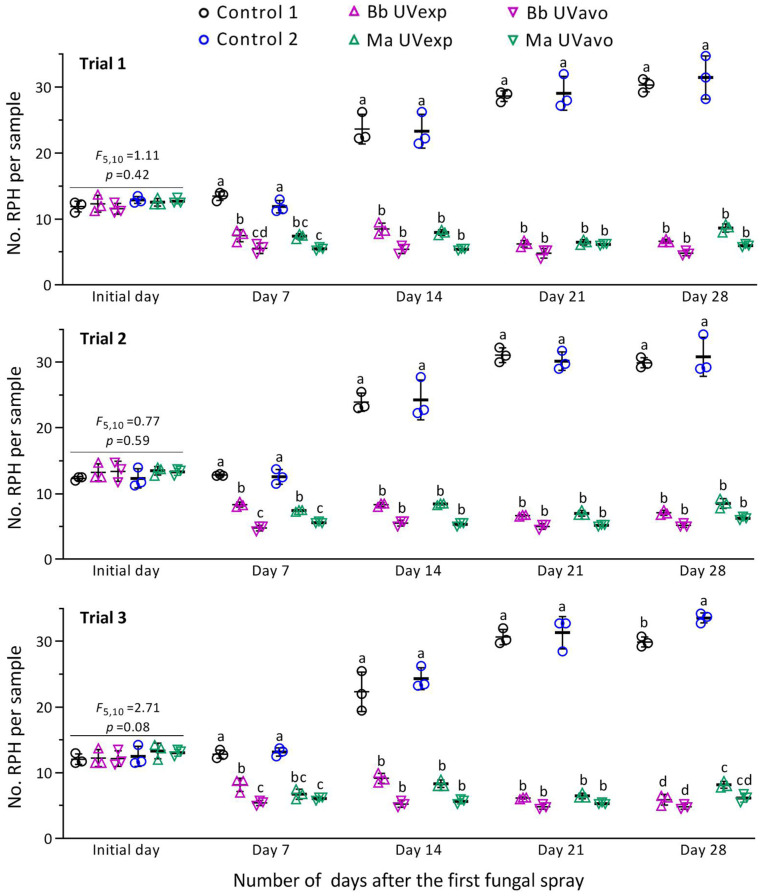
Suppression of *N. lugens*-dominated RPH population by emulsifiable oil formulations of *B. bassiana* ZJU435 (Bb) and *M. anisopliae* CQ421 (Ma) in Trials 1–3. Each formulation was sprayed twice before 10 a.m. (UVexp) or after 5:00 p.m. (UVavo) at a 14-day interval over the 4-week period of field trials. RPH density data were log_10_-transformed in one-way ANOVA. Different lowercase letters indicate significant differences (*p* < 0.05 in Tukey’s test) among the treatments of each field trial on a given day after the first spray. Error bars: SDs of the means (shown with symbols, four samples per replicate) from three replicates per treatment or control.

**Figure 4 insects-14-00307-f004:**
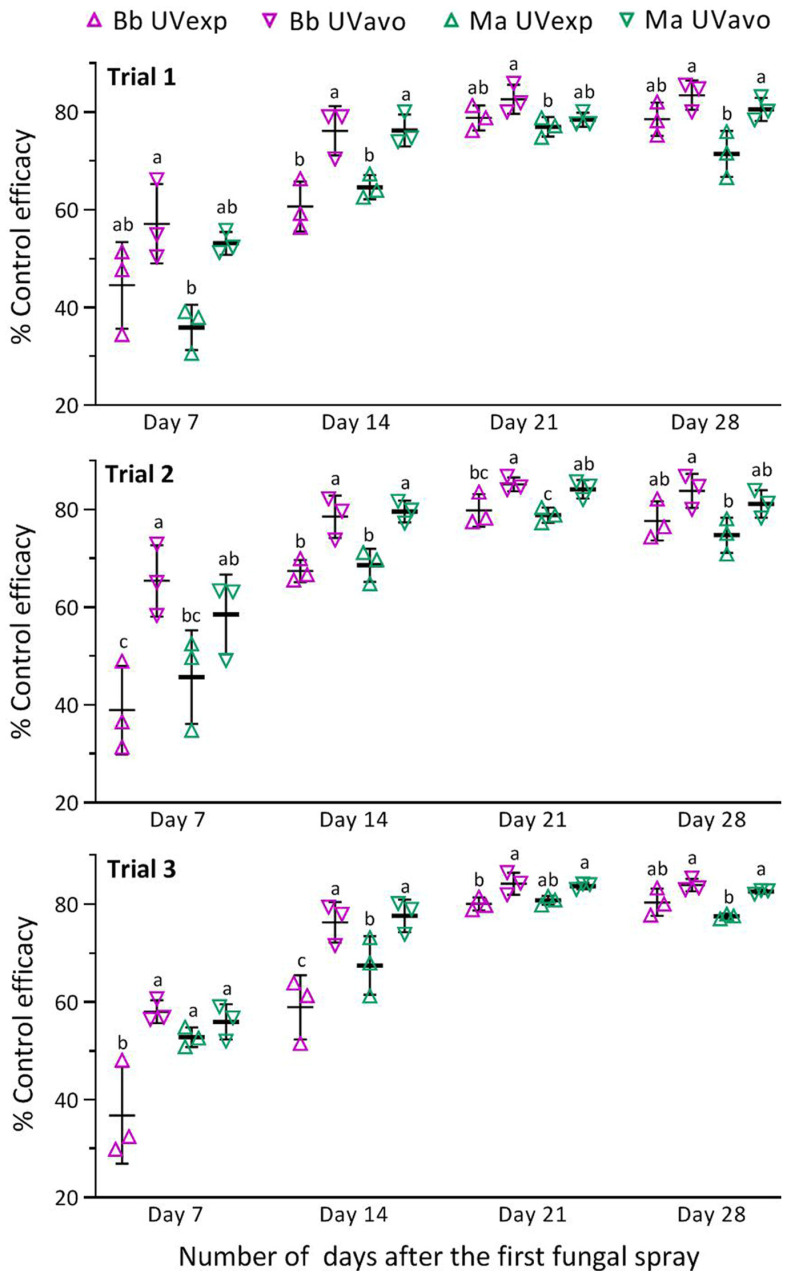
Field efficacies of RPH control by emulsifiable oil formulations of *B. bassiana* ZJU435 (Bb) and *M. anisopliae* CQ421 (Ma) in Trials 1–3. All percent data were converted to arcsine square roots in one-way ANOVA. Different lowercase letters denote significant differences (*p* < 0.05 in Tukey’s test) among the fungal treatments of each field trial on a given day after the first spray. Error bars: SDs of the means from three replicates (shown with symbols, four samples per replicate).

**Figure 5 insects-14-00307-f005:**
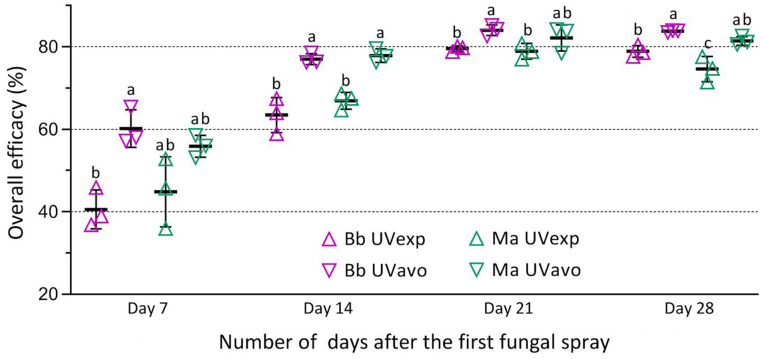
Overall comparison of RPH control efficacies between the treatments of *B. bassiana* ZJU435 (Bb) and *M. anisopliae* CQ421 (Ma) in three field trials. All percent data were converted to arcsine square roots in one-way ANOVA. Different lowercase letters denote significant differences (*p* < 0.05 in Tukey’s test) among the treatments on a given day. Error bars: SDs of the means(shown with symbols, three replicates per fungal treatment) from three field trials. The dotted lines simply favor comparisons betwee treatments over the 4-week period of field trials.

## Data Availability

All experimental data are included in this paper and Appendix A.

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
