# Peer review of "Timing of Fungal Insecticide Application to Avoid Solar Ultraviolet Irradiation Enhances Field Control of Rice Planthoppers"

_insects, 2023, doi:10.3390/insects14040307_

Round 1
Reviewer 1 Report
The authors provide important considerations for the temporal application of bioinsecticides for the control of rice planthoppers, demonstrating that the time of day at which the bioinsecticides are applied can influence the efficacy of the bioinsecticide. This is an interesting idea which could be very useful for the biological control industry. However, I think caution needs to be expressed when drawing conclusions from this work. Other factors aside from UV, including temperature/CO2 levels, are likely to differ between night and day. I therefore think there may need to be some reframing of the conclusions, to consider the fact that unless other variables (eg temperature/CO2 levels) were controlled, it is difficult to definitively conclude that the bioinsecticides perform better because of reduced UV exposure.
A recurring consideration is the use of 'fungal insecticides', I think bioinsecticide might be better here. The use of 'mycoinsecticide' is also used, so the authors should be consistent with their terminology- i'd suggest replacing fungal insecticides with bioinsecticides.
Moreover, the strain names (e.g. CQ421/ZJU435) and species names (Ma/Bb) appear to be used interchangeably in the text/figures. I would stick with the species names (Ma/Bb)- a bit less confusing for the reader, although keep the strain names in the methods section.
Abstract:
Line 27- at 14-day intervals instead of 'at a 14-day interval'
Line 29- i'm not sure this level of detail is required in the abstract, I would remove the percentages
Line 34- remove 'sunny'
Introduction:
Good background overview of the literature, with just some minor typographical suggestions.
Line 41: since the 1980s
Line 44: 'annually' instead of 'over all the year round'
Line 44: resistance instead of resistances
Line 46: remove 'easy'
Line 50: 'despite the increasing development of resistance'
Line 50: 'it is therefore a challenge to develop alternative strategies for effective RPH control'
Line 57: rice damage instead of rice damages
Line 58: 'favour environmentally friendly fungal insecticides' instead of 'choose'
Line 72: high temperatures instead of high temperature
Line 73: remove 'strong'
Line 76: 'considered' instead of 'concerned'
Line 89: 'Our aims are as follows' instead of 'our goals aims at':
Line 92: remove 'hot'
Line 92: 'fungal spraying' instead of 'fungal spray'
I would also remove the last line of the introduction
Methods
Methods are adequately explained, with just some minor typographical suggestions.
Line 107: 'Three adjacent rice-shrimp rotation paddy fields', can remove 'adjacent to one another'
Line 111: mid-summer instead of middle summer
Line 116: 'established' instead of 'set up'
Line 119: remove 'located' and change was to 'were'
Line 121: includes instead of include
Line 138: 'spraying' instead of 'spray'
Line 148: 'the sampling sites' instead of 'sample sites'
Line 150/151: remove last line of paragraph
Results
Line 174: remove 'As illustrated in figure 1', and start sentence with 'mean and maximum daily temperatures'
Line 184: remove 'In one-way ANOVA'
Line 185: In each trial instead of 'of each trial'
Line 186: remove 'an insignificant difference'
Line 187: 'although significant differences were observed between the no. of conidia of ZJU435 versus CQ421'. It would be good to say that there were significantly greater counts of conidia in Bb treatments versus Ma, then the final line of section 3.2 can be removed.
Line 191-194 is speculative, I would remove.
Figure 2: 'Field trials of B. bassiana...' below the figure could be removed. This also applies for Figure 5.
line 204- could the word 'sample' be replaced by 'plant'?
Line 208: controls instead of control
Line 229: 'high efficiencies' is a bit subjective, would be better to back these statements up with statistics, 'significantly greater efficacy of UVavo vs UVexp' etc. Do this throughout the manuscript for results which are backed up with statistics.
Figure 4- typo on the y-axis, should say 'efficacy' rather than 'effecacy'
Line 240- significantly higher in the UVavo treatment than UVexp'. Don't need to put 'as illustrated in Figure 5', just put '(Figure 5) after statement
Discussion
Discussion section is quite brief, and there is not much discussion in the context of the wider literature. I also think it is important to address the fact that other variables were not controlled for in this study (temperature/CO2), so whilst you show an indication that UV could influence biocontrol efficacy, it is difficult to conclude this without factoring in the other variables.
Line 258: control instead of controls
Line 260: Indicates rather than implicates
Line 275: remove 'that were obviously more suitable for fungal action after the second spray on 275 August 24 or 25'
Line 277: remove 'markedly'. If there was a statistical test performed, say it was significantly higher.
Reviewer 2 Report
Xu et al. present the results of a single field experiment performed across three fields to compare the effect of applying two biological insecticides in the morning or evening. They conclude that evening application provides a slightly higher level of pest control due to reduced exposure to UV radiation. The study will be of interest to readers of Insects.
The manuscript requires major modification before it is acceptable for publication.
Major points.
The field experiment was poorly replicated and has little power to detect differences between the two insecticides tested. This needs to be made clear. Currently the authors attribute this lack of power to changes in the pest population in the control plots, but this is incorrect.
The figures 3 and 4 are difficult to read and need to be improved for readability (see comments below).
The data need to be checked for normality/homoscedasticity prior to the analyses. The authors should also consider using mixed models as they used a repeated measures design on their experimental plots.
The English text needs to be checked by the MDPI production editors.
I have written numbered comments and suggestions for improvements on a scanned copy of the manuscript.
Numbered points (see scanned copy)
1. The title could be improved. I suggest "Timing of fungal insecticide application to avoid ultraviolet irradiation improves control of rice planthoppers".
2. Unnecessary text deleted.
3. Text meaning is unclear. Please reword or eliminate.
4. Delete text as already stated previously.
5. Text meaning is unclear. Please reword.
6. Please provide a reference on sensitivity of conidia to UVB.
7. Text on possible future studies in the Introduction is irrelevant. Delete.
8. Coordinates usually given as North, then East.
Please say in which year the study was performed.
10. The word "block" is used, but I think the authors mean "plot". I presume there were five experimental plots x three replicates (not mentioned) to produce a total of 15 plots per field.
Whether the plots were arranged in three BLOCKS was unclear.
Given that FIVE treatments were applied, the design involving just THREE replicates is weak and the statistical analyses will have low power. In future experiments the number of replicates should be equal or greater than the number of treatments.
11. If the experimental design was in BLOCKS there should be a block effect described in the results, or even block*treatment interactions. I think the authors used a randomized plot design. This needs to be clarified.
12a. Was a surfactant or wetter-sticker used with conidia applications? Not mentioned.
12b. the term is "performed on four fields-of-view"
12c. Please provide an estimate of the proportion of the RPH infestation that comprised N. lugens in each field (not mentioned in the Results section).
13. Text meaning is unclear. Please reword.
14. How did the authors ensure that data met the assumptions of normality and homoscedasticity required for ANOVA? You performed an experiment with a repeated measures design in which observations were taken from the same experimental plots on several occasions. The observations are therefore not independent in time. You should make this clear in the statistical section, or use mixed-models to analyze your data.
From a simple examination I would say that there were likely to be problems with heteroscedasticity in some of the data (e.g. Fig 3).
15. The axes labels of Fig. 2 require editing.
16. The fact that you could not detect a significant difference between the fungal insecticide treatments is because you used an experimental design with the smallest possible number of replicates and your ANOVA has very low power. It is nothing to do with changes in the number of insects sampled in the control treatment.
17. Again, you could not detect a significant difference between the morning and evening application treatments is because you used an experimental design with the smallest possible number of replicates and your ANOVA has very low power.
You need to make this clear.
18. Fig 3. Fig 4: Please remove triangles above columns for clarity.
19. Instead of repeating details of the experimental procedures, you should describe what the figure indicates in the title. What do white columns indicate? What do gray columns indicate? What do dark gray columns indicate? Please provide a clear key (no triangles).
20. Fig 4 is too small to read easily, please increase size. See point 18, see point 19.
21. Fig 5: Axis X label needs editing.
22. Final sentence is irrelevant. Delete.
23. All data are not in the manuscript and should be supplied as a supplemental file (e.g. Excel)
There are a few formatting errors in the references.

Round 2
Reviewer 1 Report
Thank you very much for checking the manuscript and making the changes I suggested.
In response to the use of 'fungal insecticides', I am satisfied with your explanation, and am thankful to the authors for providing me with this information. I think that the authors should therefore either use 'fungal insecticide' or 'mycoinsecticide', and be consistent with the terminology throughout the manuscript. Whichever term they prefer.
I am also happy for them to keep the percentages in the manuscript, but wonder if all percentages need to be used. As a reader, I struggle to know what to take away from the sentence, whereas the previous sentence summarises the results sufficiently, in my opinion.
I appreciate also that the study shows evidence of temporal differences in the efficacy of the fungal insecticides, and that with field trials, certain factors are hard to control. My point about the discussion I think is still important, regarding the other factors that may influence the efficacy of the biocontrol agents, and I do encourage the authors to include a sentence or two stating the limitations of the study in this context- as CO2 records aren't available, the potential effect of temperature could therefore be included. As you say, you record the temperatures at the trials, so it would be good to refer to them in this part of the discussion.
I would like to thank the authors for addressing the points I made, and accept their responses to my points.
Reviewer 2 Report
The authors have taken into account my suggestions. I only spotted the following issues that should be addressed prior to acceptance:
L4. Author name should be "Zhen-Xin Wen" correct? (not XinWen)
L102 -should read: "ha each"
L114 - should read: 2-m wide buffer between plots.
L158, L160 should read: logarithms
L176 should read: ...after Spray Application
L184, L185 should read: Bb, Ma
Figure 2, Figure 3 - -titles should mention that analyses were performed on log-transformed values.
